# communications
# engineering

# A multi-model architecture based on deep learning for aircraft load prediction

Chenxi Sun [1,2], Hongyan Li [1,2 ✉], Hongna Dui[3], Shenda Hong [4,5 ✉], Yongyue Sun[1,2], Moxian Song[1,2], Derun Cai[1,2], Baofeng Zhang[1,2], Qiang Wang[3], Yongjun Wang[3] & Bo Liu[1,2]

Monitoring aircraft structural health with changing loads is critical in aviation and aerospace engineering. However, the load equation needs to be calibrated by ground testing which is costly, and inefficient. Here, we report a general deep learning-based aircraft load model for strain prediction and load model calibration through a two-phase process. First, we identified the causality between key flight parameters and strains. The prediction equation was then integrated into the monitoring process to build a more general load model for load coefficients calibration. This model achieves a 97.16% prediction accuracy and 99.49% goodness-of-fit for a prototype system with 2 million collected flight recording data. This model reduces the effort of ground tests and provides more accurate load prediction with adapted aircraft parameters.

[1] School of Intelligence Science and Technology, Peking University, Beijing, China. [2] Key Laboratory of Machine Perception (Ministry of Education), Peking University, Beijing, China. [3] The Aviation Industry Corporation of China, Ltd., Chengdu Aircraft Design&Research Institute, Chengdu, China. [4] National Institute of Health Data Science, Peking University, Beijing, China. [5] Institute of Medical Technology, Health Science Center of Peking University, Beijing, China . ✉email: leehy@pku.edu.cn; hongshenda@pku.edu.cn

Aircraft structural health monitoring is of great significance to prevent aircraft damage, avoid air crashes and promote the development of aviation and aerospace engineering: Aircraft with high cost, complex structure, and safety requirements must have high-reliability[1]. With the maturity of safety and design theories of electrical appliances, accidents caused by the failure of aircraft electronic components are decreasing, while faults of the aircraft structure system, especially fatigue failure, become increasingly prominent. Once fatigue damage occurs, it often leads to air crashes and casualties[2].

Structural load analysis is a key technique of in-service aircraft life monitoring and structural health management[3]. In flights from takeoff to landing, the aircraft structures are constantly subjected to alternating loads. Statistics indicate that the load is the major cause of fatigue failure, accounting for >50% of the total mechanical structure failures[4]. At present, this problem has not been well solved in engineering because load sensors are usually not available to be equipped on the aircraft in service due to the restriction of cost, reliability, and maintenance[5].

In this situation, in-service aircraft structural load ($F$) is usually identified by the on-board flight parameters ($X$) based on the load model $F = f(X)$[6]. However, such a load model needs to be calibrated by the ground test, including wind tunnel test and computational fluid dynamics analysis, which has a heavy workload, a long cycle, and the risk of accidental damage to the aircraft[7]. Thus, in current engineering practice, the model of one aircraft is used for the whole fleet (Fig. 1b), i.e., a general load model. But this model is not so general as only the data of one aircraft is used. Subtle differences in structure and abrasion among aircrafts will weaken their reliability. Establishing the load model adapt to every aircraft is more accurate, but also unrealistic and expensive.

Early classical approaches could establish equations suitable for different aircrafts[8] (Fig. 1a), where the load ($F$) was calculated from the measured strains ($E$) based on the strain-load linear regression equation $F = kE + b$[9], where parameters $k, b$ are weight and bias. They mainly depended on the strain gauges pasted on the main load-transferred path of each aircraft. But the strain gauges gave the risk of falling off, data drift and missing[10]. The data error could be up to 40% when the operational demand is high. Although many practices have improved the measurement accuracy[11,12], operation and maintenance would require regular expensive tests, and once the strain gauge pasted inside the structure fails, it can hardly be compensated. Thus, the cumbersome engineering process hinders the establishment of the general load model. (Section S2 of the Supplementary Information covers additional related research.)

In this paper, we propose a two-phase prediction process to get a general aircraft load model as shown in Fig. 1c (Details are in Fig. S1): (I) predicting strains from flight parameters, (II) calibrating strains and obtaining load. Compared with the end-to-end method from flight parameters to load (Fig. 1b), it can create a general model for the fleet by calibrating the strain-load equation; Compared with the method of directly using the strain-load equation (Fig. 1a), it can avoid strain gauges that could fail at any time, which is meaningful for the long-term use of an aircraft. The following findings were made while using our method:

Finding 1: There is a deep learning-based Granger causality (our previous work[13]) from flight parameters to strains, including the relation among products of flight parameters and strains.

Finding 2: The strain-load equation of one aircraft can be obtained by processing the strain-load equation of another aircraft calibrated by the ground test with a correction coefficient.

Based on Finding 1 and 2, we can design a two-phase method to achieve a more general load model, consisting of (i) predicting strains from flight parameters and (ii) calculating load from strains. i.e., $F = kE + b = kf(X) + b$. This idea keeps the classical idea of calculating load from strains to ensure the model generality and keeps the practical idea of utilizing flight parameters to ensure the model's accuracy. In application, it cuts down some engineering processes: reducing the use of unreliable strain gauges through the prediction method and avoiding potentially damaging tests by using convenient flight tests instead of cumbersome ground tests.

However, it is not easy to integrate the strain prediction method into the load monitoring process directly as the response relations between flight parameters and strains are highly complex during flight. Meanwhile, noise and electronic interference often exist[14] and small differences among flight parameters may cause large prediction differences.

Finding 3: Flight attitude always affects the result of strain prediction. In the flight course, the aircraft will have many phases like takeoff and landing, and many actions like turning and circling[15], even the same flight parameters may respond to different strains. i.e., there are different relations, causing multiple data distributions. But most individual models lack the ability to learn them all due to the premise of independent identical distribution (i.i.d)[16,17].

Finding 4: Data corruption in a short period is the main noise. But the load prediction is sensitive: A 5% error of load may cause almost 20% error in damage and life, and 90% reliability of life requires at least 97% reliability of load[7].

Based on Finding 3 and 4, the promising approach is to design a robust flight parameter processing flow and flight attitude coding method. In this way, we can build a unique deep learning model for each flight attitude to improve accuracy.

Besides, practical aviation application requires interpretable methods, but most deep learning models are uninterpretable. The 22nd article of the European Union's General Data Protection Regulation stipulates that a subject of algorithmic decisions has a right to meaningful explanation regarding said decisions[18]. Engineers always justify a result, verify and revise the method using domain knowledge familiar to them[19]. Unfortunately, the common opinion holds that deep learning models are blackboxes[20]. Through our previous investigation[21–23], we found that an expert's understanding of a method is usually based on their knowledge, i.e., which input features are important to the result, and what is the contribution of each feature to the result. Based on this, we achieved the interpretation method by finding the key flight parameters and their contributions to load prediction, which is a preliminary exploration to explain the deep learning method in aviation engineering.

Finding 5: There are the important flight parameters in the load prediction process, namely normal overload, attack angle, inner aileron deflection, Mach number, and barometric altitude. This interpretation is important to the model expansion, data analysis and accuracy improvement of our model.

The major advantages of our study are fourfold: (1) We proposed a deep learning method for strain prediction, which is a multi-model responding to 36 defined hybrid flight attitudes. The model can achieve 97.16% average prediction accuracy. The error is less than the 5% tolerance in the aviation industry. (2) We proposed a coefficient calibration method to achieve a more general load model without increasing costs. It can achieve 99.49% goodness-of-fit. In this way, the model generality can be improved by the data from more aircrafts, where only one cumbersome ground test is needed and the others are replaced with convenient flight tests. (3) We designed a key feature-based method and a alternative model-based nonredundant multiple tree to interpret deep learning models. It showed the important flight parameters in the load prediction process. (4) Our tests are implemented on real flight records. We have collected 2,003,159 flight records from 5 aircrafts. Each record consists of 28 flight

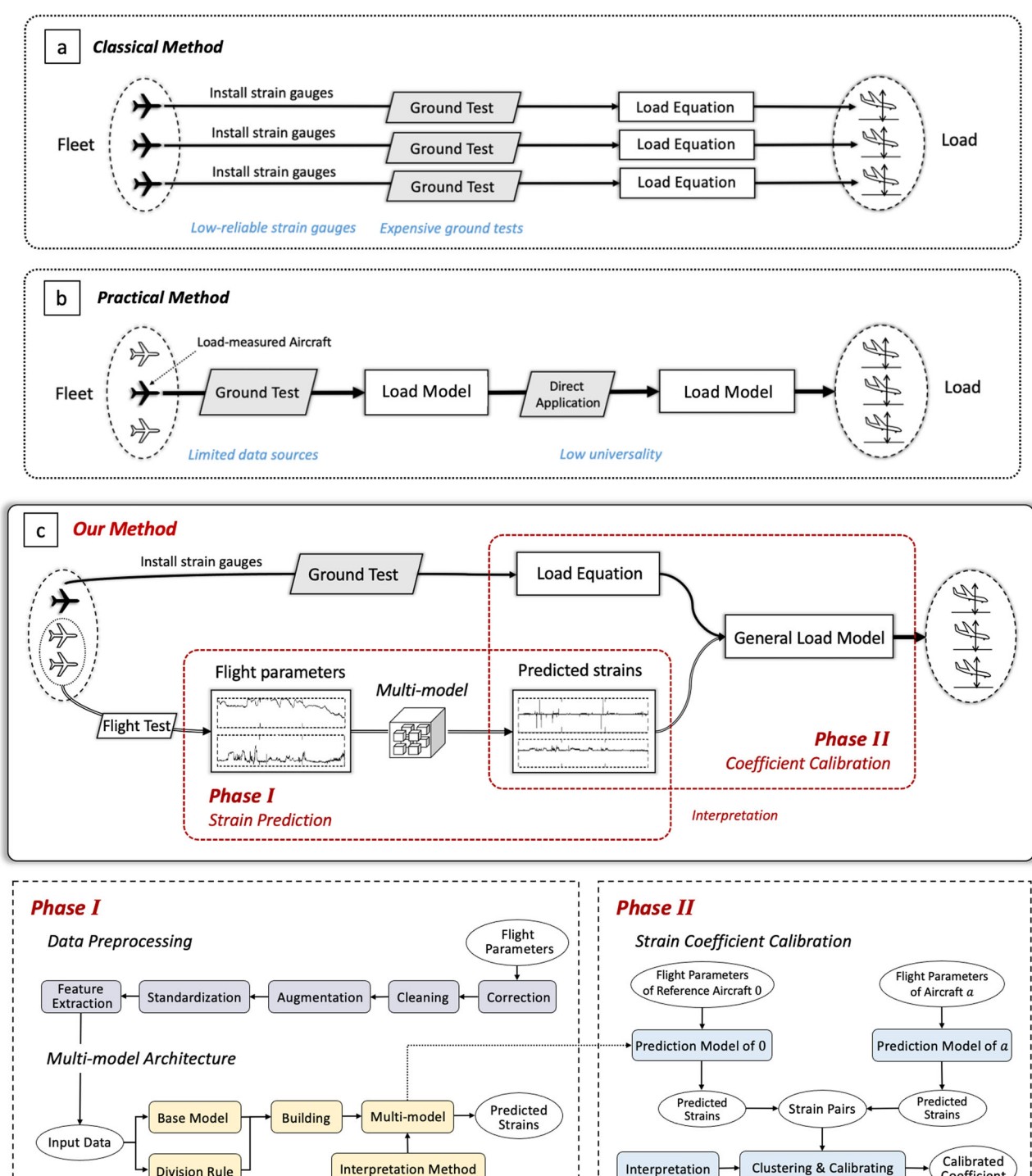

**Fig. 1 Two-phase aircraft load prediction process with deep learning. a** Classical method: Establishing the load equations (from strains to load) for each aircraft in the fleet. **b** Practical method: Using the load model (from flight parameters to load) of one aircraft as the general load model of the fleet. **c** Our method: Developing the general load model of the fleet through deep learning-based two-phase process: (i) strain prediction from flight parameters and (ii) coefficient calibration for load model.

parameters and 10 strains. The real experimental data and environment ensure the reliability of our method.

## Results

**The deep learning-based two-phase prediction method addressed most challenges in aircraft load prediction.** We make best use of the advantages of practical methods (flight parameter-load model)

and classical methods (strain-load equation)[24–30], and propose a two-phase process for load prediction as shown in Fig. 1c:

Phase I: We establish the strain-load equation $F^0 = kE^0 + b$ for the reference measured-load aircraft 0 through the ground test (solid aircraft icon), and build flight parameter-strain model $E^a = f(X^a)$ for other aircrafts $a$ through flight tests.

Phase II: We calibrate the coefficient $SF^a$ between the strain $E^0$ of the reference measured-load aircraft 0 and the strain $E^a$ of the

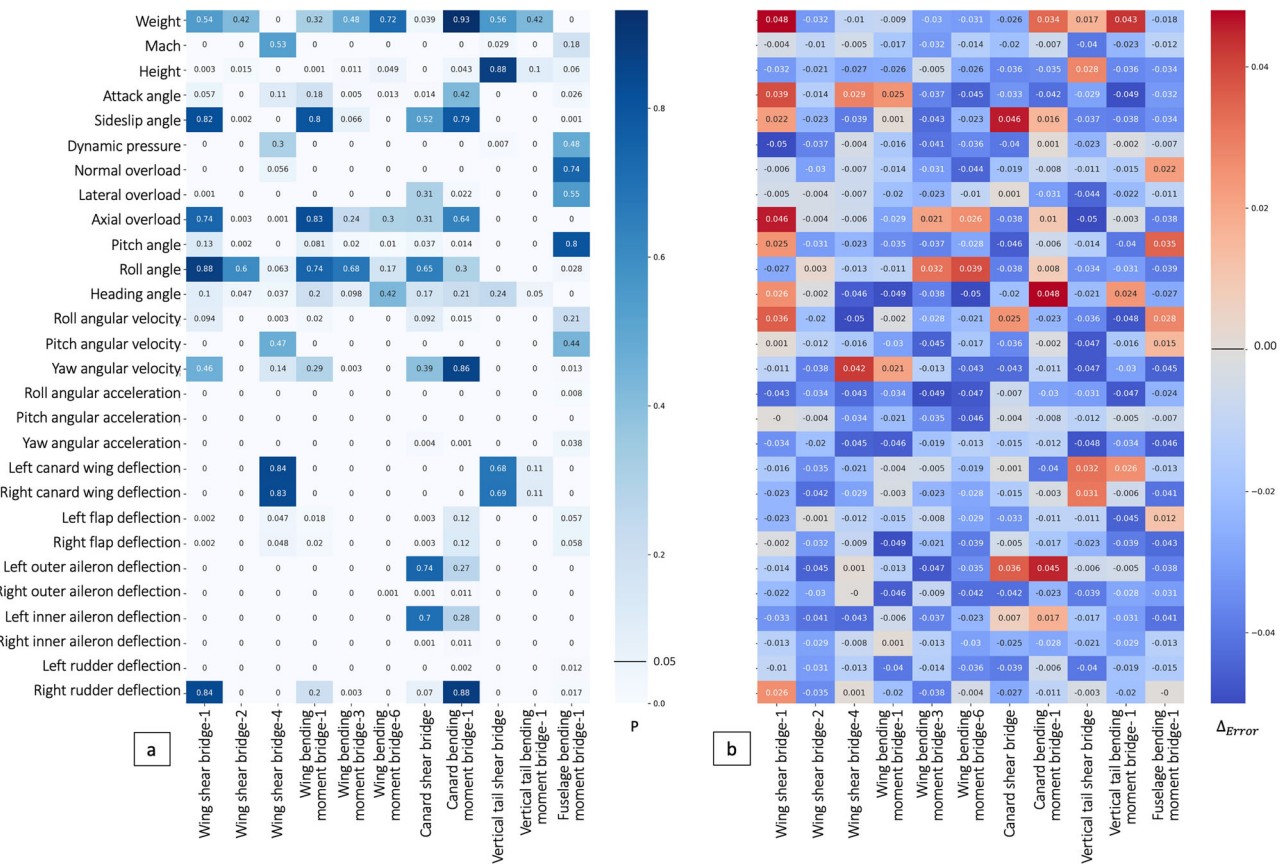

**Fig. 2 Granger causality from flight parameters to strains. a** Classical Granger causality test (have causality if $P < 0.05$). **b** Deep learning-based Granger causality (have causality if $\Delta_{Error} < 0$). The causal relationship found by our method is basically consistent with that found by the classical method.

aircraft $a$, $E^0 = SF^a E^a + b^a$. Finally, the load model of each aircraft $a$ is $F^a = k(SF^a f(X^a) + b^a) + b$.

To reduce the cumbersome aerospace engineering process, our method only needs one ground test and does not require strain gauges in later practice; To model complex response relations among flight data, our method creates a load model for each aircraft and adopts the multi-model architecture based on flight attitudes; To improve the practicability of the model, our method is equipped with interpretation methods for deep learning corresponding to each phase.

In the following text, variables with superscript 0 belong to the reference measured-load aircraft, while variables with superscript $a$ belong to other aircrafts.

**Performance of the strain prediction in Phase I**. The average accuracy of our model is 97.16%. All prediction accuracy can basically be stable at 97%. The error is less than the 5% tolerance in the aviation industry.

Result 1: Flight parameters and strains have deep learning-based Granger causality, laying the foundation for the prediction model. The previous work is based on the prior knowledge that the load has relations to strains or flight parameters. Inspired by this, we explored the relationship between strains and flight parameters. In this way, strains can be obtained by the prediction from flight parameters rather than the measurement from strain gauges, thus avoiding data errors over time.

The classical Granger causality is a statistical hypothesis testing method that determines whether one time series is the cause of another. The feasibility of the prediction method can be validated by testing the causality between flight parameters and strain. But flight parameters are multivariate time series from complex

system, which makes the test difficult to implement: The classical Granger causality only analyzes two variables but multiple variables; The prior knowledge assumes that the relation between variables is linear and can not analyze the complex nonlinear dependency; It only examines static causality and ignores dynamic causality.

Thus, to solve these issues, the deep learning-based Granger causality is proposed in our previous work, which measures causality between two variables through a deep learning model: Deep neural networks can model complex nonlinear relationships without prior knowledge; Joint modeling reduces the spatial complexity of the model from $O(n^2)$ to $O(n)$; The correlation time periods can analyze the causality that dynamically change over time.

We concluded: There is the Granger causality from flight parameters to strains! As shown in Fig. 2a, more than 70% pairs have the classical Granger causality. The p-value of the F-test is <5%; As shown in Fig. 2b, more than 80% pairs have the deep learning-based Granger causality (our previous work[13]). Almost all flight parameters can help deep neural networks to forecast strains after adding them to input (Table S4–S6 contain more detailed results.). This finding indicates that the strains can be predicted from flight parameters, especially using deep learning methods! Then, by combining the equation between strains and load, we can use flight parameters to predict the load indirectly.

Result 2: Flight parameters mainly have data corruptions in the short period. Their products are conducive to deep learning models with additional physical meanings. Flight parameters are flight data recorded by the flight recording system, including about 30 parameters such as weight, normal overload, pitch angle, etc. Due to noise and electronic interference, the collected data is not clean.

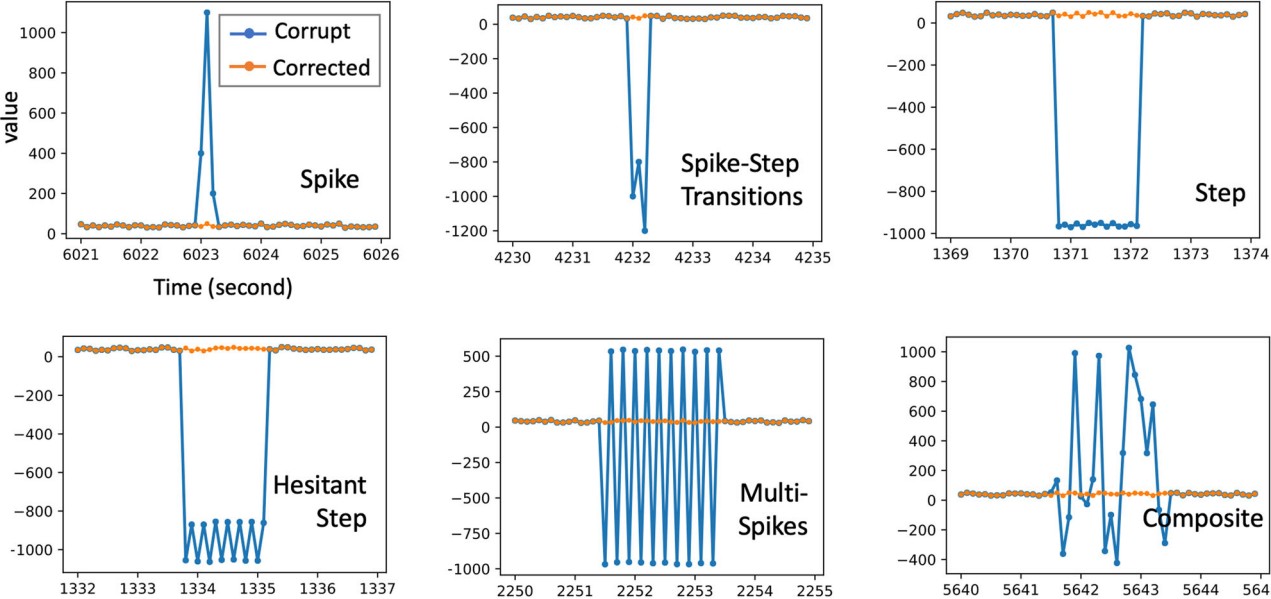

**Fig. 3 Data correction for flight parameters.** The main data corruptions are spikes, steps, and composite.

As shown in Fig. 3, the main problem is data corruption in the short period, which is mainly reflected in spikes, steps, and composite. After eliminating these three types of data corruption, the strain prediction accuracy based on the deep learning model was improved by about 5% (Figs. S5 and S6 contain more detailed results). Meanwhile, outliers that can be detected by the angle-based method[31] exist when regarding all signals as a multi-dimensional sequence. And the redundant features will affect the prediction accuracy. The feature can also be extracted from the frequency domain, we created additional features by fusing in different frequency domains[32] to enrich data. After the above processing for flight parameters, the strain prediction accuracy was improved by about 2% (Tables S7–S9 contain more detailed results.). We also found that 75.6% (3258 in $11 \times C_{28}^2$) pairs of flight parameters' products and strains have the deep learning-based Granger causality. And the product of some flight parameters may have a additional physical meaning, e.g., the lift is equal to the normal overload multiplied by the weight. Thus, we extend $n$ flight parameters to $C_n^2 + n$ features. It improved prediction accuracy by about 6%.

Result 3: There are different relations between flight parameters and strains under different flight attitudes. The hybrid strategy based on the maneuver code and the PITS rule shows a good result of relation division. Flight parameters not only reflect the load but also reflect the change of flight attitude. In the flight course, the aircraft will go through multiple phases, including takeoff, cruise, landing, etc. And in one phase, the aircraft may also perform some flight actions. In different flight attitudes, the relation between flight parameters and strains may different. If we use all the data to train a deep learning model without considering the aircraft attitude, the average prediction accuracy is <90%. Therefore, it is necessary to first divide the entire data set into multiple subsets, and then design a specific model for each subset. We have tested the impact of different division methods on the prediction, and finally determined the best division strategy, which is based on the flight phase and action, specifically, a hybrid method of maneuver and point-in-the-sky (PITS).

Our previous work coded the maneuver[15]. It divided the flight records by the identified flight actions (The first table in Fig. 4a). We subdivided the maneuver into 28 categories (Code M01-M28 are described in Table S10), which can improve the prediction

accuracy to nearly 95%. Meanwhile, in our experience of strength testing, the range of flight parameters can also reflect the aircraft's state. Our previous work defined the PITS, dividing the flight records into 540 subsets by the reference value of flight parameters (height, Mach number, weight, normal overload) (The second table in Fig. 4a). But it has hardly improved the prediction accuracy, even reduced to 85%. This is because some subsets have too little data, which hinders the training of the deep learning model. This under-fitting issue also occurs in maneuver division.

As shown in Fig. 4b, different division methods lead to different accuracy. Finally, we get a hybrid coding strategy, namely, maneuver C1-C9, PITS F1-F4 (The third table in Fig. 4a). That is, the final strategy divided the original dataset into $9 \times 4 = 36$ subsets. 9 maneuvers consist of turn/circle, pull rod/ push up, dive turn, jump turn, split-s, loop, half loop, roll, and ground attack. 4 PITS consist of $H < 5000$ and $Nz < 3.0$, $H < 5000$ and $Nz \leq 3.0$, $H \leq 5000$ and $Nz < 3.0$, $H \leq 5000$ and $Nz \leq 3.0$, based on the height ($H$) of 5000$m$ and the normal overload ($Nz$) of 3.0$g$ (Refer to Fig. S7).

Result 4: Multi-model architecture is more accurate and more robust for strain prediction as it can eliminate the influence of different flight attitudes on flight parameters. Compared with strains, flight parameters are more reliable, easy to obtain, and low-cost[24]. They represent flight status, attitude and working status[33]. We proposed a deep learning multi-model to predict strains from flight parameters. In detail, we first divide the whole dataset into multiple subsets, then we design specific deep neural networks for each subset, and finally, we get a structure consisting of multiple models.

We introduce the specific model based on Multi-Layer Perceptron (MLP) to learn each subset. MLP is a kind of deep neural network, forming a highly complex nonlinear dynamic learning system based on a complex network structure formed by many simple processing units of neurons widely connecting with each other. In the training process, we also integrate some optimization methods of feedback mechanism, model uncertainty evaluation, neural architecture search, parameter update strategy, etc. Note that after dataset division, some subsets inevitably are a small sample. For example, some flight actions like split-s have a short duration, yielding a dataset many orders of magnitude

**a**

Maneuver

| No. | Maneuver type | Category | Code |
|---|---|---|---|
| 1 | Ture or circle | C1 | M01, M02, M03, M04, M05, M06 |
| 2 | Push pod and pull up | C2 | M07, M08 |
| 3 | Dive turn | C3 | M09, M10, M11 |
| 4 | Jump turn | C4 | M12, M12, M14 |
| 5 | Split-S | C5 | M15, M16, M17, M18 |
| 6 | Loop | C6 | M19, M20, M21 |
| 7 | Half loop | C7 | M22, M23 |
| 8 | Roll | C8 | M24, M25, M26, M27 |
| 9 | Ground attack | C9 | M28 |

Point-in-the-sky (PITS)

| Height | Mach | Weight | Normal overload |
|---|---|---|---|
| H≤2E3 | Ma<0.85 (Ma<0.9) | W<2E4 | Nz≤-2 |
| 2E3<H≤5E3 | 0.85<Ma≤1.15 (0.9<Ma≤1.1) | 2E4<W≤2.5E4 | -2<Nz≤0 |
| 5E3<H≤8E3 | 1.15<Ma (1.1<Ma) | 2.5E4<W≤3E4 | 1<Nz≤3 |
| 8E3<H≤1.1E4 | | 3E4<W≤4E4 | 3<Nz≤5 |
| 1.1E4<H≤1.4E4 | | 4E4<W | 5<Nz≤7 |
| 1.4E4<H | | | 7<Nz |

| No. | Category | Type |
|---|---|---|
| 1 | F1 | H≤5E3, Nz≤3 |
| 2 | F2 | H≤5E3, 3<Nz |
| 3 | F3 | 5E3<H, Nz≤3 |
| 4 | F4 | 5E3<H, 3<Nz |

Data Division by Hybrid Strategy

| No. | Category | No. | Category | No. | Category | No. | Category |
|---|---|---|---|---|---|---|---|
| 1 | C1 F1 | 10 | C1 F1 | 19 | C1 F1 | 28 | C1 F1 |
| 2 | C2 F1 | 11 | C2 F1 | 20 | C2 F1 | 29 | C2 F1 |
| 3 | C3 F1 | 12 | C3 F1 | 21 | C3 F1 | 30 | C3 F1 |
| 4 | C4 F1 | 13 | C4 F1 | 22 | C4 F1 | 31 | C4 F1 |
| 5 | C5 F1 | 14 | C5 F1 | 23 | C5 F1 | 32 | C5 F1 |
| 6 | C6 F1 | 15 | C6 F1 | 24 | C6 F1 | 33 | C6 F1 |
| 7 | C7 F1 | 16 | C7 F1 | 25 | C7 F1 | 34 | C7 F1 |
| 8 | C8 F1 | 17 | C8 F1 | 26 | C8 F1 | 35 | C8 F1 |
| 9 | C9 F1 | 18 | C9 F1 | 27 | C9 F1 | 36 | C9 F1 |

Data Clusters after Division by Hybrid Strategy

36 Clusters

**b**

**Fig. 4 Construction and model selection rules of deep learning-based multi-model. a** Data division strategy. **b** Prediction accuracy under different data division strategies. The hybrid division method yields the highest accuracy.

smaller than those typically used in deep learning applications. As shown in Fig. 5c, small sample makes the accuracy of MLP low. Thus, we also introduce the Ridge Regression (RR) and Light Gradient Boosting Machine (LightGBM).

Three individual basic machine learning methods achieve different accuracy and the data size affects their performance as shown in Fig. 5b and c. Our multi-model can select the best basic model through an accuracy feedback mechanism. Therefore, regardless of the amount of data, its performance is always the upper bound of the performances of RR, LightGBM, and MLP. Meanwhile, the multi-model has different prediction accuracy when using different data division methods: As shown in Fig. 5d, the hybrid method is the best. It indicates that both the maneuver and the numerical ranges of flight parameters will affect the strain response. Besides, as shown in Fig. 5a, our method is more accurate and has stronger generalization: Although the accuracy of the classical end-to-end method is adequate, our method is more accurate (blue bar); Our method performs better on new data as we take uncertainty into account during training, whereas the accuracy of other methods decreases (orange bar). Our method has transfer learning mechanisms that can be fine-tuned to adapt to new data (yellow bar).

In summary, as shown in Fig. 5e and f, our method achieved an accuracy of over 95% on different strains and aircrafts (Table S11, Figs. S8 and S9 contain more detailed results).

**Performance of the coefficient calibration in Phase II.** We perform coefficient calibration of all strains of each aircraft in the fleet with the reference measured-load aircraft to form the personalized load model of each aircraft. The method can achieve 99.49% goodness-of-fit for strain coefficient calibration on average.

Result 1: Prediction model helps to find stain pairs to be calibrated with low computational complexity and avoid the gap among the flight parameter distributions of aircrafts. Before calibrating, in the original flight data of two aircrafts, it is unknown which two strains correspond. Finding strain pairs through the similarity requires traversing the entire dataset and has the complexity $\mathcal{O}(n^2)$, where $n$ is the number of flight records and usually millions. Besides, this method is not always reliable because of alignment errors caused by the differences in data distribution of different aircrafts.

To reduce the algorithm complexity and avoid the distribution gap, we propose a data pair construction method based on the

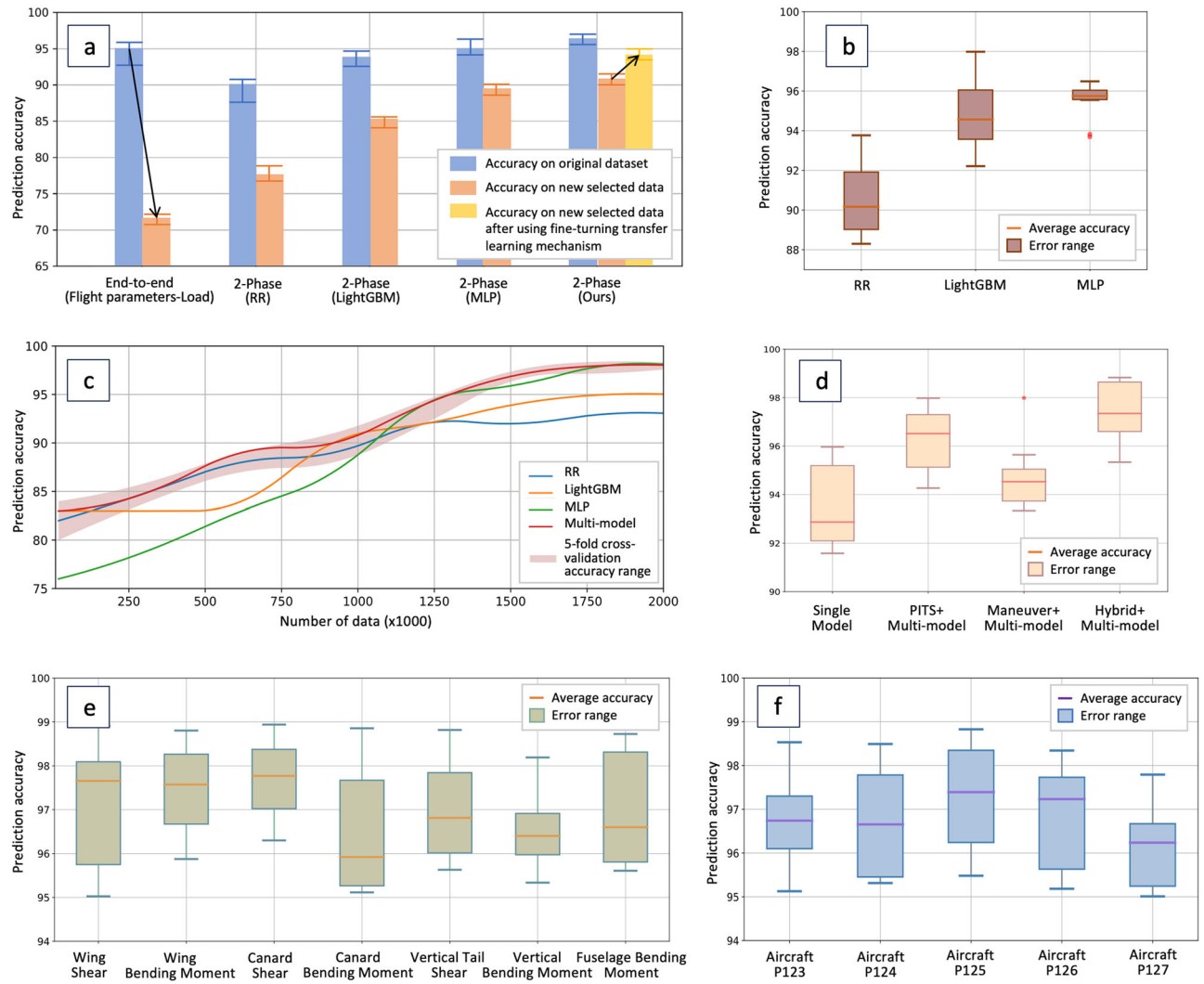

**Fig. 5 Prediction performance of proposed deep learning multi-model and comparison with baselines. a** Prediction accuracy of different baselines. Black arrows indicate that the accuracy of the classical method for new data is significantly reduced, but our method uses a transfer learning machine to improve the generalization. **b** Prediction accuracy of different base models. MLP model yields the highest accuracy. **c** Change of Prediction accuracy with data volume. Small sample scenarios are not friendly to deep learning models. **d** Prediction accuracy using different model structures. Multi-model structure yields the highest accuracy. **e** Prediction accuracy of our method for 7 different strains. All predictions are >95% accurate. **f** Prediction accuracy of our method for 5 different aircrafts. All predictions are >95% accurate.

prediction model. We use the model of the aircraft 0 to predict the strain $E^0 = f^0(X^a)$ from the flight parameters of the aircraft $a$. Combining with the corresponding real strain $\hat{E}^a$ of $a$, we can get pair $(E^0, \hat{E}^a)$. Then we also use the model of $a$ to predict the strain $E^a$ from flight parameters of 0 and get $(\hat{E}^0, E^a)$. Finally, we integrate them to get the pair dataset $(E^0, E^a)$ (Fig. S10 shows the detailed process). The complexity will be reduced from $\mathcal{O}(n^2)$ to $\mathcal{O}(n)$.

Result 2: Clustering-based coefficient calibration can modify the load model for each aircraft and finally achieve a more general and low-cost load model. To revise the strain-load equation of each aircraft, we calibrate the strain coefficients between the aircraft $a$ and the reference measured-load aircraft 0 and create their relation $E^0 = SF^a E^a + b^a$.

To correct factor $SF$, our method iterated the feasible space of $SF$. In this process, as shown in Fig. 6a, the intercept $b$ under the current $SF$ was clustered based on the distribution-based method and the density-based method. At the same time, as shown in Fig. 6b, the clustering silhouette coefficient $S$ and the coefficient of determination $R^2$ were obtained and feedback to continue to

iterate and adjust $SF$. The distribution-based method models $b$ as a Gaussian distribution $\mathcal{N}(\mu, \sigma^2)$ and divides the $\sigma$ interval equally to get clusters. Then, the density-based method merges clusters with a small sample, especially at the edge of the distribution.

The method can achieve 99.49% goodness-of-fit for strain coefficient calibration (The detailed calibration results and performances are in Table S12, S13, Fig. S11 and S12). Without clustering, $SF$ may meet the regular requirement of $b$. But it results in the high local $R^2$ while the low overall $R^2$. The clustering method can alleviate this problem.

**Performance of interpretation methods.** The interpretation of the method can promote the implementation of the application. In this work, we propose the key feature-based and the alternative model-based interpretation method for deep learning models: We find the important input features to explain the prediction results and design a tree-like if-else rule to explain the calibration process.

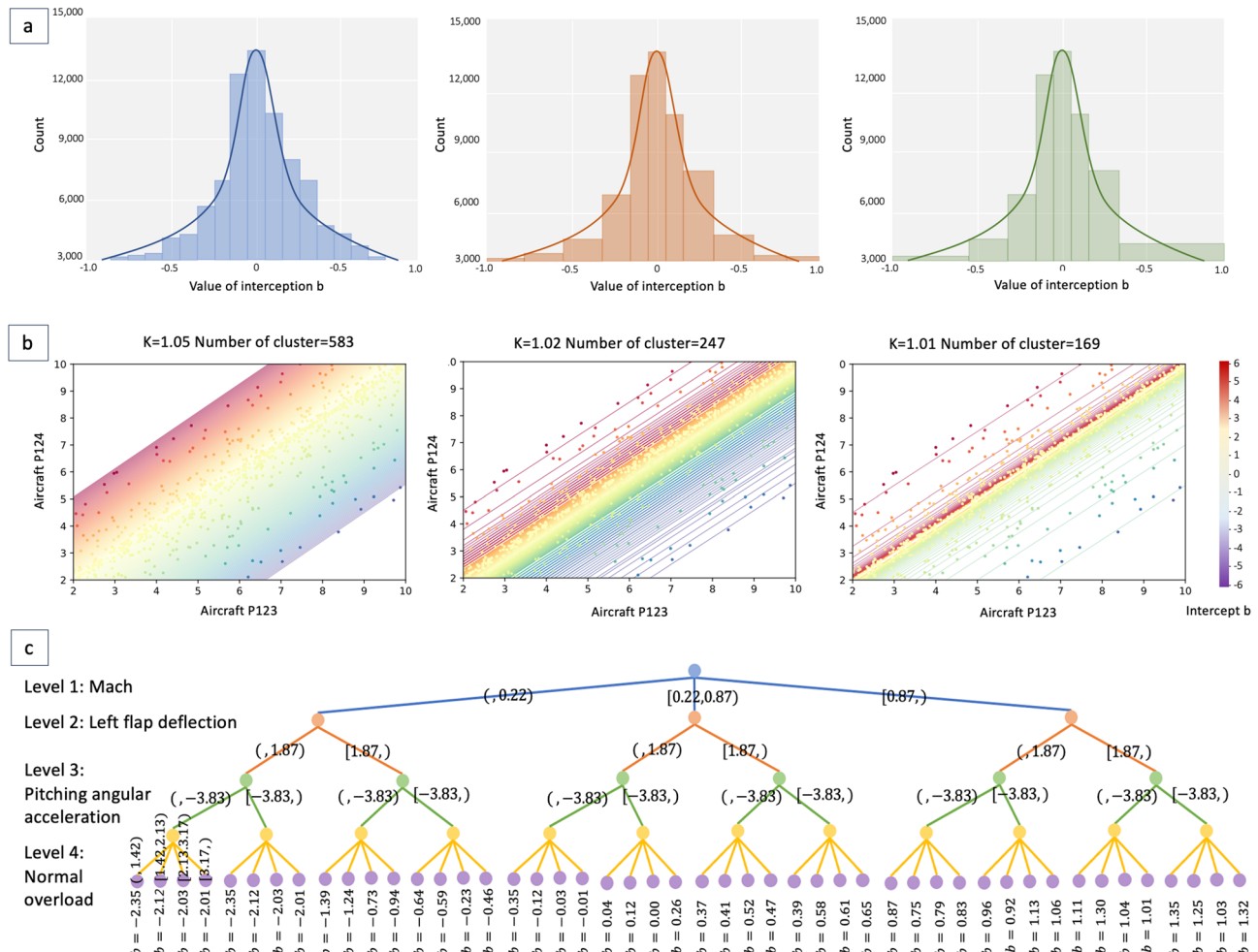

**Fig. 6 Cases of coefficient calibration. a** Clustering case: change of clustering. The clusters are defined by the value range of interception b in Eq.(10). Our method gradually enlarges the interval to achieve the appropriate clustering. **b** Calibration case: change of correction factor. The number of correction coefficients changes with the interval in (**a**). **c** Interpretation case: nonredundant multiple tree (k = 1.01).

Result 1: The key feature-based interpretation method for deep learning models reveals the important flight parameters for load prediction. For the uninterpretable neural network MLP, we interpret its results by the key features of model perception, not the model itself. The interpretation is flight parameters that have a great impact on the strain prediction results. We use the SHapley Additive explanation (SHAP) method[34]: Each kind of flight parameter is calculated to get a Shapley value, which is the average contribution of a feature to the prediction in all possible coalitions. According to ranking the features of the Shapley value, we can get the important flight parameters when predicting strains. Figure 7c shows the top 20 single flight parameters and product flight parameters (The overall rankings are detailed in Table S14 and Fig. S13). For example, the five most important flight parameters are normal overload, attack angle, inner aileron deflection, Mach, and barometric altitude (height).

We also found that the important flight parameters in deep learning are different from that in the classical multiple linear regression analysis. As shown in Fig. 7a and b, the classical important flight parameters are about deflections. But deep learning models focus more on normal overload, attack angle, Mach number, and height while paying attention to deflections. This finding could provide a horizon for engineering research. The perception principles of the deep learning model at different aircraft structural components are different, although it is for the

same strain. But perception principles for strains at the same components are similar.

Result 2: The substitution model-based interpretation method gives the rules of coefficient calibration. For strain coefficient calibration, because it's a repeated clustering process, the explanation for every step is lengthy. Thus, we design the alternative model-based method. Under the same *SF*, the samples with the same intercept *b* are considered in the same class. We design a classification model to learn the relation between the flight parameters and the intercept classes. The decision tree is naturally interpretable. But it is usually the binary tree, and the same classification feature will appear in different layers, resulting in multiple paths that can reach the same class. Thus we propose the Nonredundant Multiple Tree (NMT) and prove that there is an equivalent NMT without information loss for the full binary tree (Fig. S14). The interpretation path is shown in Fig. 6c, four key classification characteristics are obtained: Mach, left flap deflection, pitching angular acceleration, and normal overload (Table S15).

## Discussion
Artificial intelligence algorithms have the potential to solve or optimize most problems in the aviation field. Flight data is large and readily available. Fusing the flight big data into prognostic information and automated decisions leads to improvements in aircraft

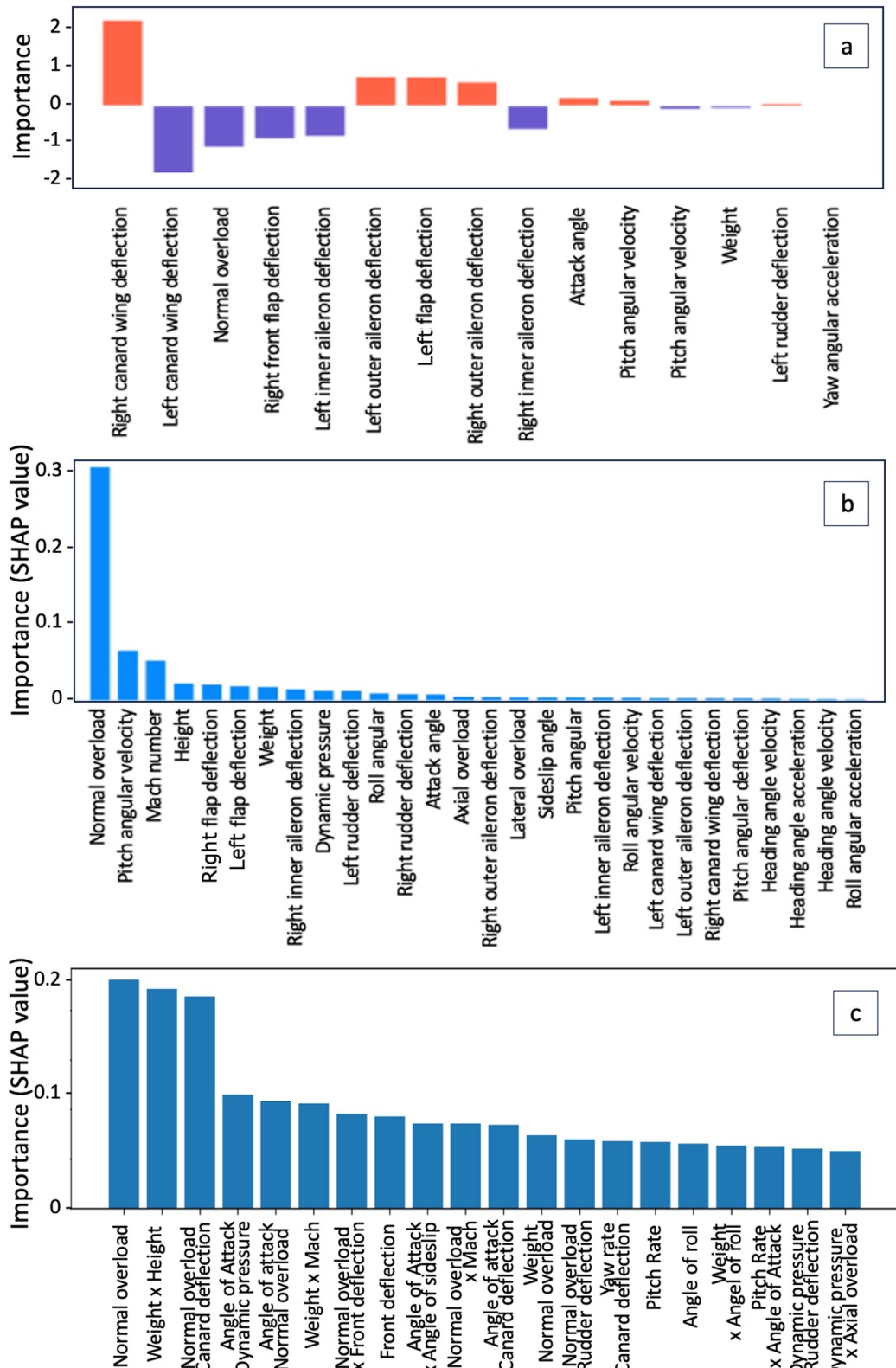

**Fig. 7 Important flight parameters for wing shear prediction. a** Importance ranking of flight parameters in the linear regression model. **b** Importance ranking of flight parameters in multi-layer perceptron model. **c** Importance ranking of extended flight parameters in multi-layer perceptron model.

Management, Affordability, Availability, Airworthiness, and Performance (MAAAP). Most aviation sub-field is full of flight data. In addition to aircraft structural fatigue research, other applications, such as aircraft anomaly diagnosis and automatic management of flight control, also contain a large amount of data, which can be implemented or improved by artificial intelligence methods[35].

Software algorithms can make up for the shortage of hardware. Deep learning[36] have achieved great success in many fields such

as medicine and industry[37]. In this work, we showed that using only one cumbersome ground test can form a mode general load model. The data-driven approach can learn the potential data relations and avoid the use of low-precision devices. In detail, we can use deep learning models to predict strains, rather than using unreliable gauges to measure strains.

Data-driven algorithms may also find the physical relation that omits in the basic physics research. For example, in subsonic flight, the existing work does not take dynamic pressure as an important characteristic for calculating the load, but our data-driven method finds that the combined characteristics, such as dynamic pressure multiplied attack angle, have a great impact on the load.

A large number of flight records provide the possibility for the application of deep learning and bring complex characteristics. Big data will improve the accuracy and reliability of the algorithm. Deep learning models require cleaned and corrected flight records. In this work, we have collected over 4 million flight records and designed a data preprocessing method to support the training of deep learning models. However, due to complex engineering processes and confidentiality and privacy reasons, data is not easy to obtain. Future work could focus on data generation, few-Shot learning, and transfer learning.

Multi-model architecture can alleviate the problem of complex data distribution in many industrial scenarios. Flight data has multiple different internal relations and feature dependence. For example, the strain response in the split-s process is twice the value of that in the turning process, even though they have the same flight parameter value. A single model is hard to model all the distributions simultaneously. Future work can apply transfer learning technology to reduce the complexity of training and construction for multi-model.

Interpretability of deep learning is one of the key issues to achieving human trust and inserting the deep learning algorithm into engineering workflow. DL models are often considered to be black-box because they typically have high-dimensional nonlinear operations, many model parameters, and complex model architectures, which makes them difficult for a human to understand.

Although interpretability is still an issue that has not been solved in the field of deep learning, we can bypass the direct interpretation of the neural network, and design some auxiliary methods. For example, in this work, there is no need to fully explain the model mechanism, and the interpretation effect can be achieved only by providing key features. Meanwhile, the simple model is easy to explain. We can use natural interpretable models to fit the results of complex models. In our work, we use a simple tree model to explain complex clustering and get good results. That is, although the general interpretation of the deep learning model is unknown, designing the specific interpretation methods for the specific problem is acceptable, and even leads to additional findings. For example, we ranked the important flight parameters when the deep learning model predicts trains and have found the five most important features, which are diffident from that in the classical analysis approach. It shows that deep learning models have a distinctive perception principle, rather than based on common physical relationships. It may provide ideas for further physical and engineering research.

**Challenges and opportunities for future work.** Compared to directly collecting strains and computing the strain-load equation, our two-phase method predicts strains using flight parameters, avoiding the problem of strain being expensive and prone to failure during flight. It is particularly meaningful for the long-term use of an aircraft, but an additional step in the process may introduce additional prediction errors. Although our method achieves acceptable errors in the aviation industry, it still falls short of the precise strain gauges used in the early stages of an

aircraft's life. Currently, our practice is to use sensing strains when the strain gauges are still reliable and to use the prediction method when they fail. To essentially avoid relying on strain gauges that could fail at any time, our future work will focus on improving the adaptability of our two-phase prediction method. At present, we use the fine-turning transfer learning mechanism in our method, which makes the model more robust to new data. Considering new scenarios and aircraft models, our future work will embed few-shot learning and federated learning technologies to make the load model not only applicable to the current fleet but also to other fleets.

## Methods

As shown in Fig. 1, building a general load model mainly involves the following processes: flight parameter preprocessing, strain prediction, coefficient calibration, and method interpretation. Refer to Supplementary Information Section S1 for detailed methods and more experiments. Notations are summarized in Table S1.

**The general load model.** First, the strain-load equation of the reference measured-load aircraft 0 is obtained by the ground test (Eq. (1)) . Then, strain prediction models of all aircrafts is built by deep learning models (Eq. (2)). And the strain coefficients between aircraft 0 and $a$ is calibrated by clustering methods (Eq. (3)). Finally, we can get the load model for all aircrafts in a fleet (Eq. (4)).

$$F = \sum_i k_i^0 \cdot E_i^0 + b^0 \tag{1}$$

$$E^{a/0} = f^{a/0}(X^{a/0}) \tag{2}$$

$$E^0 = SF^a E^a + b^a \tag{3}$$

$$F = \sum_i k_i^0 \cdot (SF_i^a E_i^a + b_i^a) + b^0 = \sum_i \alpha_i E_i^a + B = \sum_i \alpha_i \cdot f_i^a(X^a) + B \tag{4}$$

**Data preprocessing.** The dataset contains 2,003,159 records from 5 aircrafts, about 400,000 records per aircraft. Each record consists of 28 kinds of flight parameters and 10 kinds of strains (Table S2 and S3 contain data statistics, and Fig. S2 depicts a data case). We process the original flight data in five steps: (1) We filter the time series of flight parameters and strains with 8Hz stopband cut-off frequency. (2) We use the angle-based outlier detector[31] to eliminate outliers. (3) We use the Pearson correlation coefficient to evaluate the correlation among flight parameters, use multicollinearity analysis and principal component analysis[38] to remove redundant features. (4) We extract and superimpose features in frequency domain to create additional features. (5) We extend 28 flight parameters to $C_{28}^2 + 28 = 406$ input features.

**Deep learning-based granger causality.** We proposed a deep Learning-based Granger causality test (Fig. S3, S4): If the model prediction error $\mathcal{E}$ of forecasting strain $E$ by using both $E$ and flight parameter $X$ as input is less than that of forecasting $E$ by using only $E$ as input (Eq. (5)), we will conclude that deep learning-based Granger causality exists from $X$ to $E$. The prediction model is Long Short-Term Memory (LSTM)[39], presented in Eq. (6). LSTM is a variant of Recurrent Neural Networks (RNNs) that is adept at solving long-term dependency problems. In a RNN model, the current state $h_t$ is affected by the previous state $h_{t-1}$ and the current input $x_t$, $h_t = \sigma(Wx_t + Uh_{t-1} + b)$, where $\sigma$ is an activation function, and $W, U, b$ are learnable parameters. In LSTM, $f_t, i_t, o_t$ represent forget, input, and output gates, respectively. The gate utilizes the sigmoid function $\sigma$ to make the output value between $(0, 1)$, representing a certain proportion of historical information passing through.

$$\Delta_{\mathcal{E}} = \mathcal{E}(LSTM(E, X), \hat{E}) - \mathcal{E}(LSTM(E), \hat{E}) < 0 \tag{5}$$

$$
\begin{array}{lll}
i_t = \sigma(W_i x_t + U_i h_{t-1} + b_i) & \text{Input gate} & \\
f_t = \sigma(W_f x_t + U_f h_{t-1} + b_f) & \text{Forget gate} & \\
o_t = \sigma(W_o x_t + U_o h_{t-1} + b_o) & \text{Output gate} & \\
\widetilde{c}_t = \tanh(W_c x_t + U_c h_{t-1} + b_c) & \text{Candidate memory} & (6) \\
c_t = f_t \cdot c_{t-1} + i_t \cdot \widetilde{c}_t & \text{Current memory} & \\
h_t = o_t \tanh(c_t) & \text{Current hidden state} &
\end{array}
$$

**Deep learning-based multi-model architecture for strain prediction.** The original flight parameters dataset is divided into $9 \times 4 = 36$ subsets by 9 maneuver categories and 4 PITS sets with height ($H$) of $5000m$ and normal overload ($Nz$) of $3.0g$.

We build 9 Multi-Layer Perceptrons (MLP) for these 9 subsets. Each MLP contains an input layer, hidden layers, and an output layer. States are transferred by weighting between adjacent layers. Nonlinear activation occurs on neurons in the

hidden layer. The state $o_j^J$ in $J$-th hidden layer is the transformation of all states $\{o_i^I\}_{i=1}^{N_I}$ in $I$-th hidden layer $o_j^I = \sigma(\sum_{i=1}^{N_I} W_{ij}o_i^I + b_{ij})$, $N_I$ is the number of neurons in $I$-th layer. We use mean square error $L_{mse}$ (Eq. (8)) and model uncertainty loss $L_{uncertainty}$ (Eq. (9)) as the minimum bi-objective $\mathcal{L}_{MLP}$ (Eq. (7)), where $\gamma_1, \gamma_2$ are weight coefficients, $f$ is MLP model, $\theta, \mathcal{D}$ are model parameters and parameter distribution, $\mathcal{H}$ is the entropy of the predictive distribution. Using the dual objective as the loss function can increase the stability and generalization ability of our model. Meanwhile, each MLP has different model structures, reflected in 2 structural hyper-parameters and 5 training hyper-parameters, which are searched by the method of neural architecture search[40]. We also introduce ridge regression and light gradient boosting machine[41] for small subsets.

$$\mathcal{L}_{MLP} = \gamma_1 \mathcal{L}_{mse} + \gamma_2 \mathcal{L}_{uncertainty} \tag{7}$$

$$L_{mse} = \sum(E - f(X))^2 + \lambda \parallel W \parallel_2^2 \tag{8}$$

$$L_{uncertainty} = \underbrace{\mathcal{H}[\mathbb{E}_{P(\theta|\mathcal{D})}[P(f(X)|X, \theta)]]}_{TotalUncertainty} - \underbrace{\mathbb{E}_{P(\theta|\mathcal{D})}[\mathcal{H}[P(f(X)|X, \theta)]]}_{DataUncertainty} \tag{9}$$

**Clustering-based coefficient calibration for load model**. We calibrate the strain coefficients (Eq. (10)) between aircraft $a$ and reference measured-load aircraft 0 by assuming that the coefficient between the strain $E^0$ and $E^a$ is $SF^a$.

$$E^0 = SF^a E^a + b^a \tag{10}$$

To calibrate $SF$, we first proposed a prediction-based method to find the corresponding strain pairs $(E^0, E^a)$. We use the model of the aircraft 0 to predict the strain $E^0$ from flight parameters of aircraft $a$, $E^0 = f^a(X^a)$. Combining with the corresponding real strain $\hat{E}^a$ of aircraft $a$, we can get pair $(E^0, \hat{E}^a)$. In the same way, we got $(\hat{E}^a, E^a)$. We integrate them to get the pair dataset $(E^0, E^a)$. Then, we design an iterative and feedback process to iterate the feasible space of $SF$. The intercept $b$ under the current $SF$ is clustered based on the distribution-based method and the density-based method. The clustering silhouette coefficient $S$ (Eq. (11)) and the coefficient of determination $R^2$ (Eq. (12)) are obtained and feedback to continue to iterate and adjust $SF$, where $D_{intra}, D_{inter}$ means the distance within a cluster or between clusters, $D$ is the Euclidean distance, and $C$ is the cluster center.

$$S = \frac{1}{|b|}\sum_{i=1}^{|b|} \frac{D_{intra}(b_i) - D_{inter}(b_i)}{Max\{D_{intra}(b_i), D_{inter}(b_i)\}} \tag{11}$$

$$D_{intra}(b_i) = Min\, Avg\, D(b_i, \{C_{\sim b_i}\}), \quad D_{inter}(b_i) = Avg\, D(b_i, b_j \in C_{b_i})$$

$$R^2 = 1 - \frac{\sum_n (E_n - \overline{E})^2}{\sum_n (\hat{E}_n - \overline{E})^2} \tag{12}$$

Distribution-based clustering models $b$ as a Gaussian distribution $\mathcal{N}(\mu, \sigma^2)$ and divides the $\sigma$ interval equally and get clusters. Density-based spatial clustering[42] merges clusters with a small number of samples, especially those at the edge of the distribution.

**Interpretation method for deep learning models**. We use the SHapley Additive exPlanation (SHAP) method[34] to interpret MLP $f$. Each kind of flight parameter is calculated to get a Shapley value (Equation (13)), which is the average contribution of a feature to the prediction in all possible coalitions, where $z$ is the coalition vector and $M$ is the coalition size; We interpret LightGBM through the information gain Gain calculated by information entropy Ent of the training dataset $D$ (Eq. (14)), where $x_v$ means possible discrete value of $x_i$; We interpret RR by its independent variable coefficient $\beta$ (Eq. (15)).

$$SHAP(f, x_i) = \sum_{z \subseteq X\setminus\{x_i\}} \frac{|z|!(M - |z| - 1)}{M!}(f(X) - f(z)) \tag{13}$$

$$Gain(D, x_i) = Ent(D) - \sum_{x_v \in x_i} \frac{|D^v|}{|D|}Ent(D^v), D^v = \{x|x = x_v, x \in D\} \tag{14}$$

$$E = \beta_0 + \beta_1 x_1 + ... + \beta_n x_n + \mu \tag{15}$$

We design the alternative model-based method to explain how to determine the specific $b$ under the current calibration with $SF$. Because the tree model can be explained by if-else rules, we propose a Nonredundant Multiple Tree (NMT) as the alternative model. We have proved that there is an equivalent NMT without information loss for the full binary tree generated by discretization of continuous features, merging, and pruning. Assuming that the original tree is a full binary tree with $N$-layer continuous features and $K$ features are used. The NMT will have $K + 1$ layers and $\frac{2^{N-1}+1}{K+1}$ branches in each layer, the final number of category leaves is $(\frac{2^{N-1}+1}{K+1})^k$. Therefore, in this study, we only need to satisfy $(\frac{2^{N-1}+1}{K+1})^k \geq 2^{N-1}$ to construct an equivalent nonredundant multiple tree. When $N = 5, k > 1$ can satisfy $(\frac{15}{k} + 1)^k - 16 \geq 0$. Under the same $SF$, the samples with the same intercept $b$ are regarded as in the same class. We used NMT to learn the relation between flight parameters and intercept classes. In this way, the bifurcation principle of each layer

of NMT can be used to point out the characteristics that affect the bias $b$ under the same coefficient $SF$.

## Data availability

Our tests are implemented on real flight records. We have collected 2,003,159 flight records from five aircrafts. Each record consists of 28 flight parameters and 10 strains. We provide flight records of an aircraft during a flight on https://github.com/SCXsunchenxi/LoadPrediction. Supplementary Information is provided. Correspondence and requests for materials should be addressed to Chenxi Sun (sun_chenxi@pku.edu.cn), Hongyan Li(leehy@pku.edu.cn), and Shenda Hong(hongshenda@pku.edu.cn).

## Code availability

The relevant code is publicly available on https://github.com/SCXsunchenxi/LoadPrediction.

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

## Acknowledgements

This work was supported by the National Natural Science Foundation of China (No.62172018, No.62102008).

## Author contributions

H.L. managed the project. C.S., H.D and S.H led the research and development efforts. C.S., Y.S., M.S, D.C, B.Z, Q.W., Y.W., and B.L. participated in the research and the development of the prototype system. H.D., Q.W. and Y.W. collected the flight data. C.S. and H.L. wrote and revised the paper.

## Competing interests

The authors declare no competing interests.
