## [Peer Review File · Communications Engineering]

A Multi-model Architecture Based on Deep Learning for Aircraft Load PredictionReviewers' comments:

Reviewer #1 (Remarks to the Author):

The authors proposed a two-phase process for predicting aircraft load with deep learning. They used the flight parameter and strain data of in-service aircraft fleet and got high-accuracy prediction results. The work is interesting and seems suitable for engineering applications. However, the reviewer suggests the publication of this manuscript after the following revisions or illustrations.

Major revision:

1. Personally, the two-phase process may have error accumulation, Why doesn't even basic measurement error accumulate from the results?

2. Although traditional load identification has a certain error, it only needs to be calibrated on the ground through the load-strain equation. The proposed two-phase process requires more flight data (including load, strain, flight parameters) for training, which seems difficult to achieve in many application scenarios. Please further elaborate on the applicability and limitations of the methods proposed in the article

Minor revision:

1. There are many citations errors of figures, such as "Figure 2" should be modified as "Figure 3".

2. In Figure 3a Table 1, each maneuver type may be divided into multiple codes. How to divide them? Why?

3. At the first line of Page 10, there are division errors for 4 flight points.

4. At the same location, there is a concept error. $N_z=3$ means $N_z=3g$, but not $N_z=3m/s^2$

Reviewer #2 (Remarks to the Author):

In this paper, a novel approach is introduced to predict the aircraft load via using two-phase prediction process. The authors propose a multi-model architecture based on deep learning for load prediction, and remarkably demonstrate the effectiveness of the prototype system via high accuracy results compared to collected flight record datasets. The system encompasses various components, including a data preprocessing method, a flight attitude coding rule, a deep learning-based Granger causality test, and interpretability methods, all of which have contributed to uncovering new insights into flight parameters. It is impressive to witness the successful application of such a comprehensive system in aeronautics, which has the potential to impact the conventional research paradigm. Therefore, I suggest publishing the article after

implementing a few minor revisions

1. Given the complexity of the overall system, it would be beneficial to include an architecture diagram in the main text to clarify the role of each method in the two-phase process. Although I noticed that such a diagram is provided in the appendix (Fig. 7), I recommend integrating it with Fig. 1 and relocating it to the main text instead of the appendix.

2. Several symbols and representations are present in the paper, but some are inadequately defined. The method and related works are elucidated in the appendix; however, symbols used in the main text need to be explained in detail. For instance, in Eq. 3, the meaning of \mathcal{H} remains unclear. It is recommended that the author carefully examines all equation symbols and provides appropriate explanations for each of them. A possible solution would be to include a notation table to explicitly describe the variables.

3. The authors should explain why the focus is on investigating causal relationships rather than correlation (Page 6, Result 1), and highlight the specific advantages of the proposed deep learning-based Granger causality method over previous correlation analysis methods. Additionally, it would be helpful to know the overall advantages of the system proposed in this manuscript. I suggest the author provides further explanation or presents additional comparisons between the predictions of the current approach and conventional ones.

4. Though the accuracy of the predictions is well presented in Figure 3 and 4. It would be nice to see the direct predictions of time history of forces or loads (compared to the data from the flight test) to show practical application potential of the proposed approach.

5. Why design an explanation method NMT during the Phase II calibration process? Is it possible to apply the method for inference in the application, in addition to explaining the calibration rules?

6. It would be helpful to provide any relevant experiments that demonstrate the advantages of utilizing a dual objective approach that considers uncertainty (located at the bottom of page 17) in improving model generalization.

Response to Reviewer #1

Dear Reviewer,

We are extremely grateful for your review of the manuscript and appreciate your encouragement and valuable suggestions. You have raised a number of important issues. We agree with your comments and have modified our manuscript accordingly. Below we give a point-by-point response to your concerns and suggestions.

1. *Personally, the two-phase process may have error accumulation, why doesn't even basic measurement error accumulate from the results?*

Although traditional load identification has a certain error, it only needs to be calibrated on the ground through the load-strain equation. The proposed two-phase process requires more flight data (including load, strain, flight parameters) for training, which seems difficult to achieve in many application scenarios. Please further elaborate on the applicability and limitations of the methods proposed in the article.

Response: We appreciate your thorough advice. In this revision, we have added more discussions about challenges and opportunities for future work (Page 17, Line 9-23).

Challenges and opportunities for future work. Compared to directly collecting strains and computing the strain-load equation, our two-phase method predicts strains using flight parameters, avoiding the problem of strain being expensive and prone to failure during flight. It is particularly meaningful for the long-term use of an aircraft, but an additional step in the process may introduce additional prediction errors. Although our method achieves acceptable errors in the aviation industry, it still falls short of the precise strain gauges used in the early stages of an aircraft's life. Currently, our practice is to use sensing strains when the strain gauges are still reliable and to use the prediction method when they fail. To essentially avoid relying on strain gauges that could fail at any time, our future work will focus on improving the adaptability of our two-phase prediction method. At present, we use the fine-tuning transfer learning mechanism in our method, which makes the model more robust to new data. Considering new scenarios and aircraft models, our future work will embed few-shot learning and federated learning technologies to make the load model not only applicable to the current fleet but also to other fleets.

9
10
11
12
13
14
15
16
17
18
19
20
21
22
23

Compared to directly using strain data, the two-phase process does have the accumulative error, but our method achieves acceptable errors in the aviation industry (less than 5%). Most importantly, our method is mainly to avoid strain gauges that could fail at any time, which is meaningful for the long-term use of an aircraft.

In fact, in our method, "more flight data (including load, strain, flight parameters)" are only required during the model training process; when applying, only flight parameter data is required. However, when using the "load-strain equation" method, strain data is required. As

we discussed in Introduction (Page 2, Line 29-41; Page 3, Line 1-14), this method “mainly depends on the strain gauges pasted on the main load-transferred path of each aircraft. But the strain gauges gave the risk of falling off, data drift and missing...and once the strain gauge pasted inside the structure fails, it can hardly be compensated...” Compared with the ineffective and costly strain data, flight parameters, could being collected from flight recording system, are more reliable, readily available, and low-cost.

27 In this situation, in-service aircraft structural load (F) is usually identified by the
28 on-board flight parameters (X) based on the load equation $F = f(X)$ [7]. However,
29 such a load equation needs to be calibrated by the ground test, including wind tunnel
30 test and computational fluid dynamics analysis, which has a heavy workload, a long
31 cycle, and the risk of accidental damage to the aircraft [8]. Thus, in current engineer-
32 ing practice, the equation of one aircraft is used for the whole fleet (Figure 1b), i.e.,
33 a general load model. But this model is not so general as only the data of one aircraft
34 is used. Subtle differences in structure and abrasion among aircraft will weaken their
35 reliability [9]. Establishing the load equation adapt to every aircraft is more accurate,
36 but also unrealistic and expensive.

37 Early classical approaches could establish equations suitable for different aircraft
38 [10] (Figure 1a), where the load (F) was calculated from the measured strains (E)
39 based on the strain-load linear regression equation $F = kE + b$ [11], where param-
40 eters k, b are weight and bias. They mainly depended on the strain gauges pasted
41 on the main load-transferred path of each aircraft. But the strain gauges gave the

risk of falling off, data drift and missing [12]. The data error could be up to 40%
when the operational demand is high [13]. Although many practices have improved
the measurement accuracy [14, 15], operation and maintenance would require regu-
lar expensive tests [16], and once the strain gauge pasted inside the structure fails, it
can hardly be compensated. Thus, the cumbersome engineering process hinders the
establishment of the general load model.

In this paper, we propose a two-phase prediction process to get a general air-
craft load model: (I) predicting strains from flight parameters, (II) calibrating strains
and obtaining load. Compared with the end-to-end method from flight parameters to
load (Figure 1b), it can create a general model for the fleet by calibrating the strain-
load equation; Compared with the method of directly using the strain-load equation
(Figure 1a), it can avoid strain gauges that could fail at any time, which is meaningful
for the long-term use of an aircraft. The following findings were made while using
our method:

However, although our method achieves acceptable errors in the aviation industry, it still falls short of the precise strain gauges used in the early stages of an aircraft's life. Currently, our practice is to use sensing strains when the strain gauges are still reliable and to use the prediction method when they fail. To essentially avoid relying on strain gauges that could fail at any time, our future work will focus on improving the adaptability of our two-phase prediction method. At present, we use the fine-tuning transfer learning mechanism in our method, which makes the model more robust to new data. Considering new scenarios and aircraft models, our future work will embed few-shot learning and federated learning technologies to make the load model not only applicable to the current fleet but also to other fleets.

2. There are many citations errors of figures, such as “Figure 2” should be modified as “Figure 3”.

Response: Thanks for your careful checks. In this revision, we have double-checked all citations and made sure that they matched the description (Page 10, Line 31, 41; Page 12, Line 6; Page 15, Line 36, etc.).

3. In Figure 3a Table 1, each maneuver type may be divided into multiple codes. How to divide them? Why?

Response: Thanks for your comment. Please refer to “Wang Y.J, Dong J, Liu X.D, Zhang L.X. Identification and standardization of maneuvers based upon operational flight data. Chinese Journal of aeronautics, 2015, 28(1): 133-140.” for detailed division method. The divided maneuvers and their descriptions are in Table 10 in Appendix.

Table 10 Classification of Maneuvers

S/N	Basic-maneuver	Sub-maneuver	Code
C1	Turn	Common turn	M01
		Sustained turn	M02
		Decelerated turn	M03
		Steep left and right turn	M04
		Continuous steep turn	M05
	“S” turn	M06	
C2	Pull/Push	Pull	M07
		Push	M08
3	Diving turn	20° diving and turn	M09
		30° diving and turn	M10
		45-60° diving and turn, steep dive and turn	M11
4	Pull-up turn	20° pull-up and turn	M12
		30° pull-up and turn	M13
		45 60° pull-up and turn, steep pull-up and turn	M14
C5	Split-s	Split-s, oblique split-s	M15
		Split-s with vertical half roll	M16
		Oblique split-s with vertical half roll	M17
		High-speed of split-s with vertical half roll	M18
6	Loop	Complete loop	M19
		Complete oblique loop	M20
		Oblique-loop turn	M21
7	Half-loop	Half-loop overturn	M22
		Half-loop overturn with vertical half roll	M23
8	Roll	Dual pull-up turns M24	
		45-60° pull-up and roll	M25
		30-45° diving and roll	M26
		Slow level roll, larger-radius roll	M27
9	Air-to-ground attack	Divie-attack-pull	M28

The main loads on critical structures during flight are affected by the flight attitude (bank angle, pitch angle, roll rate, etc.) and status (altitude, Mach, load factor) of aircraft, and are decided by pilot’s performing maneuvers. Usually, one maneuver starts form 1g’s level flight, and then perform a series of actions with at least one CG Nz (normal load factor at the centre

of gravity) peak, at last back to level flight again. Records at level flight status as the start and end time can be found by searching forward and backward from Nz peak. According to flow chart as shown in the following figure, maneuvers can be extracted and identified from the multi-parameters time histories.

In terms of dividing maneuvers and establishing multiple deep learning models, our previous work coded the maneuver into 28 categories, but considering the possible small sample problem, we use 9 categories via the auto-ML mechanism.

4. At the first line of Page 10, there are division errors for 4 flight points. At the same location, there is a concept error. $N_z=3$ means $N_z=3g$, but not $N_z=3m/s^2$

Response: Thanks for your careful checks and pointing this out. In this revision, we have double-checked all technical terms to ensure that they are correct (Page 10, Line 13).

Response to Reviewer #2

Dear Reviewer,

We are extremely grateful for your review of the manuscript and appreciate your encouragement and valuable suggestions. You have raised a number of important issues. We agree with your comments and have modified our manuscript accordingly. Below we give a point-by-point response to your concerns and suggestions.

- 1. Given the complexity of the overall system, it would be beneficial to include an architecture diagram in the main text to clarify the role of each method in the two-phase process. Although I noticed that such a diagram is provided in the appendix (Fig. 7), I recommend integrating it with Fig. 1 and relocating it to the main text instead of the appendix.*

Response: We think this is an excellent suggestion. According to your advice, we have expanded Fig. 1 to represent each procedure in the two-phase process using the content from Fig.7. The two new modules (Phase I Data Preprocessing, Multi-model Architecture, and Phase II Coefficient Calibration) are shown at the bottom of Fig. 1.

a. **Classical Method:** Establishing the load equations (from strains to load) for each aircraft in the fleet.

b. **Practical Method:** Using the load equation (from flight parameters to load) of one aircraft as the general load model of the fleet.

c. **Our Method:** Developing the general load model of the fleet through deep learning-based two-phase process:
(i) Strain prediction from flight parameters and (ii) Coefficient calibration for load model.

Fig. 1 Two-phase Aircraft Load Prediction Process with Deep Learning

2. Several symbols and representations are present in the paper, but some are inadequately defined. The method and related works are elucidated in the appendix; however, symbols used in the main text need to be explained in detail. For instance, in Eq. 3, the meaning of \mathcal{H} remains unclear. It is recommended that the author carefully examines all equation symbols and provides appropriate explanations for each of them. A possible solution would be to include a notation table to explicitly describe the variables.

Response: Thanks for your careful checks. In this revision, we have double-checked all equations' symbols and made sure that they were all properly explained: we described all symbols and their explanations where they first appear in the main text; And the variables are explained and described explicitly by adding a notation table (Table 1) in Appendix.

Table 1 Notations and Description

Notation	Description
X, x_i, X^a	Flight parameters, a flight parameter feature, flight parameter of aircraft a
E, E_i, E^a	Strains, a strain, strain of aircraft a
F, F^a	Load, load of aircraft a
SF, SF_i^a	Calibration coefficient, SF of E_i between a and measured-load aircraft 0
f	Prediction model
θ, β, W, b	Model parameters (parameters, coefficients, weights, bias)
\mathcal{L}, L	Loss function
\mathcal{E}	Error between prediction and ground truth
$\mathcal{H}, \mathcal{I}, IV$	Entropy, mutual information, information entropy
S	Silhouette coefficient
R^2	Coefficient of determination

3. *The authors should explain why the focus is on investigating causal relationships rather than correlation (Page 6, Result 1), and highlight the specific advantages of the proposed deep learning-based Granger causality method over previous correlation analysis methods. Additionally, it would be helpful to know the overall advantages of the system proposed in this manuscript. I suggest the author provides further explanation or presents additional comparisons between the predictions of the current approach and conventional ones.*

Response: We appreciate your thorough advice. In this revision, we have added an introduction to the reasons for designing this method (Page 6, Line 28-41) and added comparative experiments in Appendix (Fig. 9, 10, Table 5): Our method infers more accurately than the classical method (Fig. 10); Using the flight parameters we infer, and the MLP model predicts more accurately (Table 5).

28 The classical Granger causality is a statistical hypothesis testing method that
 29 determines whether one time series is the cause of another. The feasibility of the pre-
 30 diction method can be validated by testing the causality between flight parameters
 31 and strain. But flight parameters are multivariate time series from complex system,
 32 which makes the test difficult to implement: The classical Granger causality only
 33 analyzes two variables but multiple variables; The prior knowledge assumes that
 34 the relation between variables is linear and can not analyze the complex nonlinear
 35 dependency; It only examines static causality and ignores dynamic causality.

36 Thus, to solve these issues, the deep learning-based Granger causality is proposed
 37 in our previous work, which measures causality between two variables through a deep
 38 learning model: Deep neural networks can model complex nonlinear relationships
 39 without prior knowledge; Joint modeling reduces the spatial complexity of the model
 40 from $O(n^2)$ to $O(n)$; The correlation time periods can analyze the causality that
 41 dynamically change over time.

Fig. 9 Framework of Dynamic Granger Causality Analysis Method Based on Deep Learning

Fig. 10 Experimental Results of the Cyclic Granger Causality

Table 5 The Accuracy of Strain Forecasting Using Flight Parameters Inferred by the Classical Granger Causality Method and the Deep Learning-based Granger Causality Method

Aircraft	P123	P124	P125	P126	P127
Classical Granger Causality	74.53%	80.01%	82.99%	80.14%	85.45%
Deep Learning-based Granger Causality	83.23%	84.97%	83.75%	85.00%	86.34%

Finding the causal relationship between Flight parameters and strain is to determine the feasibility of using flight parameters to predict strains. Flight parameters and strain data are in sequential format. The Granger causality can determine whether one time-series/sequence is the cause of another.

But flight parameters are multivariate time series from complex system, which makes the test difficult to implement: (1) The classical Granger causality only analyzes two variables and ignores the influence among multiple variables; (2) The prior knowledge assumes that the relation between variables is linear and can not analyze the complex nonlinear dependency in system; (3) The method only analyzes static causality but omits the potential dynamic causality. Thus, to solve these issues, the deep learning-based Granger causality is proposed in our previous work, which measures causality between two variables through a deep learning model.

Our deep learning-based Granger causality test (Fig. 9) has advantages compared with the classical test: (1) The limitation of classical Granger causality, which can only analyze linear relationships, is overcome by using deep neural networks to model complex nonlinear relationships, and no prior knowledge is necessary; (2) By using joint modeling, the spatial complexity of the model can be reduced from $O(n^2)$ in existing methods to $O(n)$ while analyzing the causal relationships of multiple time series; (3) The method can add correlation time periods on the basis of the original Granger causality, enabling it to effectively analyze the causal relationships that dynamically change over time in complex systems.

4. Though the accuracy of the predictions is well presented in Figure 3 and 4. It would be nice to see the direct predictions of time history of forces or loads (compared to the data from the flight test) to show practical application potential of the proposed approach.

Response: Your suggestions will greatly help to improve the academic rigor of our paper. In this revision, we have added comparison results between our method and end-to-end method (direct predictions from flight parameters to load).

Our method is more accurate and has stronger generalization: As shown in the first figure of Fig.4, although the accuracy of the end-to-end method is adequate, our method is more accurate (blue bar); Because our model is based on deep learning and takes uncertainty into account during training, it performs better on new data, whereas the accuracy of the end-to-end model decreases significantly on new data (orange bar). Our model has a transfer learning mechanism that can be fine-tuned to adapt to new data (yellow bar).

Fig. 4 Prediction Performance of Proposed Deep Learning Multi-Model and Comparison with Baselines

In fact, as stated in Introduction (Page 2, Line 27-36), the end-to-end method cannot create a general model and is expensive: It must develop a prediction model for each aircraft, and while doing so, the load data needed for the model must be obtained through the ground test, which is costly. We recommend a two-phase method because existing work can already achieve the strain-to-load calculation formula (Page 2, Line 37-41). By calibrating the coefficient between strains, a general load model of a fleet can be achieved. However, the strain gauges gave the risk of falling off, data drift, and missing, thus, we propose to predict strain from flight parameters that are readily available.

In addition to the method introduction and experimental comparison, in this revision, we have added a brief explanation of the differences between our method and traditional methods

in the Introduction section (Page 3, Line 7-14).

27 In this situation, in-service aircraft structural load (F) is usually identified by the
28 on-board flight parameters (X) based on the load equation $F = f(X)$ [7]. However,
29 such a load equation needs to be calibrated by the ground test, including wind tunnel
30 test and computational fluid dynamics analysis, which has a heavy workload, a long
31 cycle, and the risk of accidental damage to the aircraft [8]. Thus, in current engineer-
32 ing practice, the equation of one aircraft is used for the whole fleet (Figure 1b), i.e.,
33 a general load model. But this model is not so general as only the data of one aircraft
34 is used. Subtle differences in structure and abrasion among aircraft will weaken their
35 reliability [9]. Establishing the load equation adapt to every aircraft is more accurate,
36 but also unrealistic and expensive.

37 Early classical approaches could establish equations suitable for different aircraft
38 [10] (Figure 1a), where the load (F) was calculated from the measured strains (E)
39 based on the strain-load linear regression equation $F = kE + b$ [11], where param-
40 eters k, b are weight and bias. They mainly depended on the strain gauges pasted
41 on the main load-transferred path of each aircraft. But the strain gauges gave the

risk of falling off, data drift and missing [12]. The data error could be up to 40% 1
when the operational demand is high [13]. Although many practices have improved 2
the measurement accuracy [14, 15], operation and maintenance would require regu- 3
lar expensive tests [16], and once the strain gauge pasted inside the structure fails, it 4
can hardly be compensated. Thus, the cumbersome engineering process hinders the 5
establishment of the general load model. 6

In this paper, we propose a two-phase prediction process to get a general air- 7
craft load model: (I) predicting strains from flight parameters, (II) calibrating strains 8
and obtaining load. Compared with the end-to-end method from flight parameters to 9
load (Figure 1b), it can create a general model for the fleet by calibrating the strain- 10
load equation; Compared with the method of directly using the strain-load equation 11
(Figure 1a), it can avoid strain gauges that could fail at any time, which is meaningful 12
for the long-term use of an aircraft. The following findings were made while using 13
our method: 14

5. *Why design an explanation method NMT during the Phase II calibration process? Is it possible to apply the method for inference in the application, in addition to explaining the calibration rules?*

Response: Thanks for your comment. For strain coefficient calibration, because it's a repeated clustering process, the explanation for every step is lengthy. Thus, we design the alternative model-based method. Under the same SF, the samples with the same intercept b are considered in the same class. We design a classification model to learn the relation between the flight parameters and the intercept classes. The decision tree is naturally interpretable. But it is usually the binary tree, and the same classification feature will appear in different layers, resulting in multiple paths that can reach the same class. Thus, we propose the Nonredundant Multiple Tree (NMT) and prove that there is an equivalent NMT without information loss for the full binary tree. (Page 15, Line 26-38)

In addition to explaining the calibration rules, for new data, we can use the rules found by NMT to determine the intercept b during calibration, making the calibration more accurate.

Result 2: The substitution model-based interpretation method gives the rules of coefficient calibration. For strain coefficient calibration, because it's a repeated clustering process, the explanation for every step is lengthy. Thus, we design the alternative model-based method. Under the same SF , the samples with the same intercept b are considered in the same class. We design a classification model to learn the relation between the flight parameters and the intercept classes. The decision tree is naturally interpretable. But it is usually the binary tree, and the same classification feature will appear in different layers, resulting in multiple paths that can reach the same class. Thus we propose the Nonredundant Multiple Tree (NMT) and prove that there is an equivalent NMT without information loss for the full binary tree. The interpretation path is shown in Figure 5d, four key classification characteristics are obtained: Mach, left front flap deflection, pitching angular acceleration, and normal overload.

26
27
28
29
30
31
32
33
34
35
36
37
38

6. *It would be helpful to provide any relevant experiments that demonstrate the advantages of utilizing a dual objective approach that considers uncertainty (located at the bottom of page 17) in improving model generalization.*

Response: We concur with your recommendation. Using the dual objective (Equation 3) as the loss function can increase the stability and generalization ability of our model.

$$\mathcal{L}_{MLP} = \gamma_1 \mathcal{L}_{mse} + \gamma_2 \mathcal{L}_{uncertainty} \quad (3)$$

$$\mathcal{L}_{mse} = \sum (E - f(X))^2 + \lambda \|W\|_2^2 \quad (4)$$

$$\mathcal{L}_{uncertainty} = \underbrace{\mathcal{H}[\mathbb{E}_{P(\theta|\mathcal{D})}[P(f(X)|X, \theta)]]}_{\text{Total Uncertainty}} - \underbrace{\mathbb{E}_{P(\theta|\mathcal{D})}[\mathcal{H}[P(f(X)|X, \theta)]]}_{\text{Data Uncertainty}} \quad (5)$$

In this revision, we have presented the new experimental results in Appendix 5.4. As shown in Figure 13, using the dual objective as the loss function can increase the stability and generalization ability of our model. And the trade-off between the two objective is set at 1:1 (the coefficient of $\mathcal{L}_{mse}=0.5$).

Fig. 13 Model Performance when Using the Dual Objective as the Loss Function

REVIEWERS' COMMENTS:

Reviewer #3 (Remarks to the Author):

The authors have addressed all of my concerns. I suggest the publication of this paper in the current form.

Reviewer #4 (Remarks to the Author):

The authors have adequately addressed all of the comments, and in my opinion, this manuscript is suitable for publication.